# ACNN-BiLSTM: A Deep Learning Approach for Continuous Noninvasive Blood Pressure Measurement Using Multi-Wavelength PPG Fusion

**DOI:** 10.3390/bioengineering11040306

**Published:** 2024-03-25

**Authors:** Mou Cui, Xuhao Dong, Yan Zhuang, Shiyong Li, Shimin Yin, Zhencheng Chen, Yongbo Liang

**Affiliations:** 1School of Life and Environmental Sciences, Guilin University of Electronic Technology, Guilin 541004, China; mathew_cui@163.com (M.C.); xh.dong@siat.ac.cn (X.D.);; 2Shenzhen Institute of Advanced Technology, Chinese Academy of Sciences, Shenzhen 518055, China; 3National Supercomputing Center in Xi’an, Xi’an 710100, China; zhuangy1@126.com; 4Guangxi Key Laboratory of Metabolic Reprogramming and Intelligent Medical Engineering for Chronic Diseases, Guilin 541004, China

**Keywords:** multi-wavelength PPG, feature fusion, blood pressure prediction, deep learning

## Abstract

As an essential physiological indicator within the human body, noninvasive continuous blood pressure (BP) measurement is critical in the prevention and treatment of cardiovascular disease. However, traditional methods of blood pressure prediction using a single-wavelength Photoplethysmographic (PPG) have bottlenecks in further improving BP prediction accuracy, which limits their development in clinical application and dissemination. To this end, this study proposed a method to fuse a four-wavelength PPG and a BP prediction model based on the attention mechanism of a convolutional neural network and bidirectional long- and short-term memory (ACNN-BiLSTM). The effectiveness of a multi-wavelength PPG fusion method for blood pressure prediction was evaluated by processing PPG signals from 162 volunteers. The study compared the performance of the PPG signals with different individual wavelengths and using a multi-wavelength PPG fusion method in blood pressure prediction, assessed using mean absolute error (MAE), root mean squared error (RMSE) and AAMI-related criteria. The experimental results showed that the ACNN-BiLSTM model achieved a better MAE ± RMSE for a systolic BP and diastolic BP of 1.67 ± 5.28 and 1.15 ± 2.53 mmHg, respectively, when using the multi-wavelength PPG fusion method. As a result, the ACNN-BiLSTM blood pressure model based on multi-wavelength PPG fusion could be considered a promising method for noninvasive continuous BP measurement.

## 1. Introduction

Blood pressure (BP) is an essential physiological parameter in the human body and refers to the BP against the walls of blood vessels. BP allows us to assess the function of heart and blood vessels [1]. Maintaining normal BP levels helps reduce the risk of heart disease and prevents diseases such as atherosclerosis and stroke [2]. Hypertension is a prevalent health problem worldwide, with an estimated 1.28 billion people aged 30–79 years suffering from the disease. Hypertension has become a common chronic disease, with its prevalence increasing every year, and one of the leading causes of premature death in humans. Hypertension is also closely related to obesity, high blood cholesterol and diabetes, etc. Maintaining a high BP for a long time may cause substantial damage to the heart, kidneys and other organs and lead to myocardial infarction, stroke and other serious consequences [3]. However, about 46% of patients have no visible symptoms and do not realize they have high BP. For this reason, the World Health Organization encourages people to take regular BP measurements to note changes in BP for early prevention and treatment.

Traditional noninvasive BP measurement methods are susceptible to many limitations. For example, cuff sphygmomanometers need to be appropriately worn, need to be kept at rest during measurement, and are accessible to external interference that can affect the results. In addition, specific individuals, such as children, pregnant women and the obese may not be suitable for traditional BP measurement methods [4]. To overcome these limitations, scientists are developing more advanced noninvasive BP measurement technologies, such as contactless BP measurement and wearable BP monitors. These new technologies provide a more accurate, convenient, and continuous way to monitor BP and are expected to improve the effectiveness and convenience of BP monitoring. Ambulatory BP monitoring is a medical method of continuously measuring an individual’s BP value over 24 h. Mobile BP monitoring is achieved by wearing a portable monitor that automatically measures BP at specific intervals, including daytime and nighttime [5]. As BP changes with different activities and sleep states, ambulatory monitoring can provide more accurate and comprehensive BP data, which is more helpful in assessing a hypertension status and monitoring the effectiveness of treatment. It also reveals the fluctuation of a patient’s BP, helping doctors to more accurately formulate a personalized treatment plan and promoting the efficacy and safety of hypertension management.

A PPG is a biomedical signal that refers to blood flow pulsation signals. PPG signals can be measured using photoplethysmography sensors placed on the skin’s surface. These sensors measure the minor fluctuations caused by blood flow beneath the skin, making it an excellent physiological signal for predicting BP. In recent years, many researchers have fed PPG signals into deep-learning models to predict BP [6,7,8]. Maqsood et al. [9] compared different machine learning methods and different feature extraction techniques for BP estimation using PPG signals and concluded that deep learning algorithms can achieve better results on time domain features. Zhong [10] proposed a BP estimation model based on a hybrid attentional gating U-Net (MAGU), which utilizes PPG signals and incorporates a hybrid attentional gating mechanism to enhance feature extraction in deep networks. Bernard [11] used PPG signals for BP estimation and evaluated different machine-learning algorithms. Rong et al. [12] developed a deep learning model consisting of a signal pre-processing module and an MTFF neural network module. H. Ankishan [13] developed a mathematical model and system that uses physiological signals and voice messages to calculate systolic blood pressure (SBP) and diastolic blood pressure (DBP). Hu et al. [14] proposed a global hybrid multi-scale convolutional network to obtain multi-scale information by summing three branches over attention weights on multiple scales. Ali [15] developed a hybrid LSTM-ANN model for BP prediction and deployed it on the STM32 microcontroller for actual time prediction.

Researchers have shown an increasing interest in studying multi-wavelength photoplethysmography (MWPPG) in recent years. While a traditional single-wavelength PPG measurement primarily focuses on parameters such as BP and heart rate, an MWPPG measurement offers a wealth of additional information, including indicators of cardiac function, blood flow velocity, and vascular elasticity. The propagation speed and characteristics of a PPG in different tissues can be measured by utilizing multiple light sources with different wavelengths. The advantage of this approach lies in its ability to acquire a more comprehensive and detailed biological signal, aiding researchers in gaining a deeper understanding of cardiovascular health and the underlying mechanisms of related diseases. Sirkiä [16] developed an MWPPG sensor that combines an external pressure generation system and a sensing system to detect different skin depths and extract information about various blood vessels in the skin. Lu et al. [17] developed a method to assess systemic vascular resistance (SVR) based on the small artery pulse transit time (aPTT) of an MWPPG. Liu et al. [18] proposed a PCA-based MWPPG cuffless BP measurement algorithm. Slapničar [19] used a modified RGB camera to estimate BP by measuring the PTT between different skin layers. Jukka-Pekka Sirkiä [20] developed a device that can receive signals from different depths of the cutaneous vasculature, proposed that PPG signals with longer wavelengths are more sensitive to pressure-induced vasodilatation than PPG signals with shorter wavelengths, and emphasized the potential of the MWPPG for BP prediction.

In this paper, we combine the continuous wavelet transform (CWT) into a multi-wavelength PPG fusion algorithm to overcome the limitations of single-wavelength PPG signals in cuffless BP monitoring. After filtering the MWPPG signals and removing outliers, the pre-processed four-channel data are individually CWT transformed and added to the dataset as images. We applied the method of fusing the four-channel CWT images into a 12-channel array for inputting into the deep learning model, which achieved better results than simply feeding the CWT-transformed images of the single-wavelength PPG signal to the deep learning model. Finally, we verify the effectiveness of the proposed MWPPG fusion-based BP prediction algorithm on a self-constructed dataset and discuss the reasons why better results can be achieved.

## 2. Materials and Methods

### 2.1. Data Collection

In the experiment, we recruited volunteers for PPG signal acquisition. A self-designed multi-wavelength PPG detection device was used [21]. The acquisition wavelength and depth are shown in Figure 1. The detailed information of the monitoring device is as follows: a 200 Hz sampling rate; a DCM08 sensor module for data collection whose wavelengths are 660 nm (Channel 1), 730 nm (Channel 2), 850 nm (Channel 3) and 940 nm (Channel 4); an ADPD4100 multi-mode sensor front-end module; and a STM32F411 microprocessor module with a core of ARM Cortex-M4 of 100 MHz. The device acquires PPG signals from each volunteer’s fingertip at a sampling rate of 200 Hz, with a duration of 1 min for each acquisition, and the collected data is saved as a txt file. This study was approved by the Biomedical Ethics Committee of Guilin University of Electronic Science and Technology (GUET-20220301-001). After the volunteers signed the informed consent form, the PPG data collection started.

The data collection process for this experiment has the following requirements:Before data collection, the subject is required to clean the fingertip area to avoid contamination affecting the test results;Before data collection, the subject should sit quietly and remain calm for 15 min to ensure the accuracy of the measurement results;During data collection, the subject is to naturally extend the index finger of the left hand and lightly press it on the sensor, keeping it still;Immediately after acquiring the PPG signal, the BP of the volunteer was measured; the device used to measure BP was the Omron HEM-7201, and the estimated BP was used as a label for constructing the model.

We ultimately retained 1 min of an appropriate PPG signal from each of the 180 volunteers for the following study. Individual samples with missing or incomplete wavelength signals were discarded during the collection process to facilitate subsequent studies of the performance of MWPPG feature fusion. The final data collected for analysis included 162 volunteers, 66 of whom were diagnosed with hypertension (SBP ≥ 130 mmHg or DBP ≥ 90 mmHg) [22].

### 2.2. Data Processing

After data filtering to remove anomalous signals, a 0.5–8 Hz Butterworth second-order bandpass filter is used for filtering. This filter effectively removes shallow and high-frequency noise signals, leaving the filtered signal smoother and more stable. In practical signal processing applications, bandpass filters usually improve the signal-to-noise ratio and effectively remove interference noise.

In some previous related studies, single-channel PPGs were directly used as model inputs. However, this requires rigorous data pre-processing, such as filtering low-quality data noise and other operations. However, due to the temporal nature of the PPG, multiple noise-filtering processes may lead to the loss of beneficial information in the signal, which may negatively affect the model’s performance. Another part of the study transforms the PPG into a 2D CWT image form before using it as model input.

The wavelet basis function is defined as a translation followed by a scale transformation of the function in the basic function *Ψ*(*t*) as follows:(1)$$\psi a,b\left(t\right)=\frac{1}{\sqrt{a}}\left(\frac{1-b}{a}\right)$$

The fundamental functions a and b are both constants while *a* > 0. Suppose the constants *a* and *b* are varied continuously in a certain interval. Many fundamental functions satisfying the conditions can be obtained in that case, called wavelet bases. In any space L^2 (R), if there is a signal sequence *x*(*t*) square productive and *x*(*t*) ∈ L^2 (R), the continuous wavelet transform of the signal *x*(*t*) is given by:(2)$$WTx=\frac{1}{\sqrt{a}}\int x\left(t\right)\psi *\left(\frac{t-b}{a}\right)dt=\int x\left(t\right)\psi_{a,b}^{*}\left(t\right)dt=<x\left(t\right),\psi_{a,b}^{*}\left(t\right)>$$

There are many kinds of commonly used wavelet basis functions, and the same type of wavelet basis includes a variety of wavelet bases with different parameters. And each wavelet basis has a different effect, highlighting the importance of selecting a wavelet basis. With the increasing types of wavelet bases, the number of wavelet functions that can realize specific functions is also increasing, and researchers can choose the wavelet function that can achieve the best experimental results according to the requirements of the scientific research. After many experiments, it is found that the use of the “cgau1” wavelet basis function for the continuous wavelet transform to generate a two-dimensional scale map is the best effect, so the length of 5 s sliding 1 s each time in the window on the PPG to take the value. In batches, the continuous wavelet transform of “cgau1” is applied to the PPGs of 5 s segments.

## 3. Experience

### 3.1. Deep Feature Learning

#### 3.1.1. Local Feature Learning

Convolutional layers are inspired by the human eye’s experience in recognizing images: the human eye usually recognizes from parts to the whole, and parts of an image are not well related. Each node of the convolutional layer is connected to some of the previous layer’s nodes to learn the output image’s local features. By defining multiple convolutional kernels, multiple feature maps of the image to be processed can be obtained, where each pixel point corresponds to the output of a node in the convolutional layer. Since the filters of feature maps are shared, the convolutional layer essentially reduces the training parameters and time while the number of required feature maps can be increased. The channel attention mechanism, mainly used in Convolutional Neural Networks (CNNs), works by learning each channel’s weights to dynamically modify each channel’s importance, thus improving the network’s ability to recognize specific features. The channel attention mechanism usually consists of two steps: firstly, the importance scores of each channel are computed by introducing global information; secondly, these scores are applied to the original feature maps to re-weight the feature maps to enhance the feature channels with a higher importance.

#### 3.1.2. Global Feature Learning

Recurrent Neural Networks (RNNs) are a common choice when dealing with sequential data. RNNs are helpful for situations where the input data are of varying lengths and require global processing, such as physiological signals or voice characters. Traditional neural networks do not handle such situations well, mainly because of the substantial time dependence of the data and the need for the network to have the ability to process globally. The most commonly used recurrent neural network is the Long Short-Term Memory Neural Network (LSTM). Compared with the traditional RNN, the LSTM introduces a gating mechanism to solve the long-term dependency problem.

And in this study, a Bidirectional Long Short-Term Memory neural network (Bidirectional LSTM) was used. Unlike traditional LSTMs, Bi-LSTM can consider information in forward and backward directions when processing sequence data. A Bi-LSTM network consists of two LSTM networks: one processes the input sequence in sequential order, which is called forward LSTM, and the other processes the input sequence in backward order, which is called backward LSTM. After the input sequence passes through both forward and backward LSTMs, the output of each moment contains the integration of the information before and after that moment. The advantage of Bi-LSTM is that it can catch the dependencies between the before and after information. This is important for information understanding and feature extraction in sequence data. Since Bi-LSTM can consider both past and future information, it is able to better catch the contextual information in the sequence, which improves the representation ability of the model.

### 3.2. Feature Fusion Methods

In this paper, in order to improve the accuracy of BP prediction, we have used images generated from the continuous wavelet transform of 5 s PPG form data in an RGB three-channel form as the model’s input.

As shown in Figure 2, an innovative approach to incorporate a four-wavelength PPG is proposed. In this method, the four wavelengths of PPG data are converted into RGB images by first applying CWT processing to each wavelength of the PPG. Since the PPG data for each wavelength is converted to one RGB image, four RGB images can be obtained. The four RGB images were then combined using the stacking method. By stacking the RGB channels of the four images in the depth direction, the four three-channel data are stacked into one twelve-channel array. This stacking method keeps the information from each channel and provides more details for further network inputs. Ultimately, this twelve-channel RGB array is fed into the network as input. In this way, we could fully utilize the four wavelengths of PPG data and convert them into richer image representations to provide better features for subsequent experiments.

### 3.3. ACNN-BiLSTM BP Prediction Modeling

This paper uses the ACNN-BiLSTM (Attention-based CNN-BiLSTM) model to predict BP values based on PPG data. The structure of the model for deep learning is shown in Figure 3. The model consists of multiple independent sets of 2D CNN, Bi-LSTM and fully connected neural networks with added-channel attention mechanisms. Among them, the input of the network is the CWT image after pre-processing the PPG, and the image features are obtained after the convolution operation by multiple sets of cascaded 2D CNNs with an added-channel attention mechanism, which is then inputted into the Bi-LSTM neural network for prediction. As the Attention-based CNN-BiLSTM model adds the channel attention mechanism, it can discover the relatedness between input data channels and extract more features between channels.

This study inputs the pre-processed PPG signals into a 2D convolutional layer in an RGB form after CWT transformation. A channel attention layer is inserted between the 2D convolutional layers to enhance the extraction efficiency of channel-to-channel features. Batch normalization (BN) alleviates the gradient dispersion problem and improves the convergence speed. Then, the self-activation function is utilized to realize the nonlinear transformation of the features. This activation function can further improve the convergence speed of the network and prevent the gradient explosion using its self-normalization ability. The two-layer Bi-LSTM neural network is then utilized to achieve global feature learning through the forward and backward processing of the features. Since the BP prediction problem is a regression problem, we finally use two fully connected layers to receive the tensor from the output of the Bi-LSTM implicit layer and the output the BP prediction. In the experiment, the data was divided into two parts in the ratio of 0.8:0.2. The data in the ratio of 0.8 was used for training and 10-fold cross validation and the data in the ratio of 0.2 was used to come to the evaluation of the model. Meanwhile, we chose Adam as the optimizer for training the model, with the learning rate set to 0.001 and the loss function set to MAE.

In this experiment, the network model is trained on a deep learning workstation with the following hardware facility details: a CPU with a 14-core Intel (R) Xeon (R) CPU E5-2690 v4 @ 2.60 GHz, a GPU graphics card with a quad Nvidia Titan Xp of 12 G display memory, and a RAM with 192 GB.

### 3.4. Performance Metrics

Before evaluating the performance of the BP prediction of the PPG measured at different wavelengths, it is first necessary to determine the evaluation metrics. For the field of BP measurement, this paper refers to the standards of the Association for the Advancement of Medical Instrumentation (mean error (*ME*) < ±5 mmHg, standard deviation of Error (*SD*) < ±8 mmHg); in addition, this paper also cites other commonly used regression evaluation indexes, which are the coefficient of determination (*R*^2^), the mean absolute error (*MAE*), and the root mean squared error (*RMSE*), and the formulas for the five evaluation indexes are as follows:(3)$$R^{2}=1-\frac{\sum_{i=1}^{n}{\left(y_{pred}\left(i\right)-y_{true}\left(i\right)\right)}^{2}}{\sum_{i=1}^{n}{\left(y_{pred}\left(i\right)-E\left(y_{true}\left(i\right)\right)\right)}^{2}}$$
(4)$$ME=\frac{1}{n}\sum_{i=1}^{n}\left(y_{pred}\left(i\right)-y_{true}\left(i\right)\right)$$
(5)$$SD=\sqrt{\frac{1}{n}\sum_{n=1}^{n}{\left(y_{pred}\left(i\right)-E\left(y_{pred}\left(i\right)-y_{true}\left(i\right)\right)\right)}^{2}}$$
(6)$$MAE=\frac{1}{n}\sum_{i=1}^{n}\left|y_{pred}\left(i\right)-y_{true}\left(i\right)\right|$$
(7)$$RMSE=\sqrt{\frac{1}{n}\sum_{i=1}^{n}{\left(y_{pred}\left(i\right)-y_{true}\left(i\right)\right)}^{2}}$$

## 4. Results

The pre-processing stage of this paper takes the method of the simultaneous four-wavelength cutting of PPG signals to ensure the same number of samples within each wavelength PPG dataset. Meanwhile, to remove the influence of dataset division on the results, the same random seed is set for each wavelength PPG dataset when randomly dividing the dataset to ensure that the samples’ order in the training set and test set of each wavelength PPG dataset is the same. This treatment aims to ensure that no additional bias is introduced into the dataset during the division process, thus ensuring the representativeness and consistency of the samples in the training and test sets. The ten-fold cross-validation method is used in this paper to evaluate the BP prediction performance of individual single-wavelength PPG signals or four-wavelength PPG signals in the study.

### 4.1. Performance of BP Prediction Base on Single-Wavelength PPG

After pre-processing operations such as filtering, abnormal segment rejection and normalization, this study targets the single-wavelength PPG signal for further analysis. Firstly, the one-dimensional signal was converted into a two-dimensional image using CWT, and it was input into the ACNN-BiLSTM model in the form of an RGB three-channel. In this paper, several sets of experiments were conducted to evaluate the BP prediction performance of PPG signals at different wavelengths. The specific results are shown in Table 1. Through Table 1, we learn that, in the case of using a single wavelength, the best BP prediction is achieved using the PPG signal acquired by Channel 4. Compared with the PPG signal obtained by Channel 2 on the SBP prediction, the R^2^ improves from 0.82 to 0.89, the *MAE* decreases from 4.17 mmHg to 3.57 mmHg, the RMSE decreases from 9.78 mmHg to 7.72 mmHg, and the AAMI criteria were satisfied.

In a performance analysis of blood pressure prediction using data from different wavelength channels in the first trial, we found that data using the 940 nm wavelength (Channel 4) performed the best in predicting systolic blood pressure (SBP), with a coefficient of determination (*R*^2^) as high as 0.89, a mean absolute error (*MAE*) of 3.57 mmHg, and a root mean squared error (*RMSE*) of 7.72 mmHg, all of which were superior to the other channels and met the standards of the American Association for the Advancement of Medical Instrumentation (AAMI). In contrast, the data using the 730 nm wavelength (Channel 2) had the lowest error in predicting diastolic blood pressure (DBP), and although it did not have the highest *R*^2^ value, it had a small mean error, suggesting that it was more accurate in DBP prediction and also met the AAMI criteria. Data from other channels such as those of 660 nm (Channel 1) and 850 nm (Channel 3) performed relatively poorly in predicting blood pressure and did not meet the AAMI criteria. In summary, Channel 4 demonstrated the highest prediction accuracy and goodness of fit in this trial and was the optimal choice for predicting blood pressure.

The PPG data collected by Channel 4 using the 940 nm wavelength performed well in predicting blood pressure, which may be attributed to the superior ability of the 940 nm wavelength to penetrate skin and tissues, as well as its high absorption of hemoglobin and deoxyhemoglobin, which makes the PPG signal more sensitive to changes in blood flow dynamics. In addition, this specific wavelength may provide a better signal-to-noise ratio of the signal, helping to accurately extract the signal of blood pressure changes from background noise. Physiological and biochemical factors may also have different effects on different wavelengths of light, and the 940 nm wavelength may find the optimal balance among these factors.

### 4.2. Performance of BP Prediction Base on Four-Wavelength PPG

As shown in Figure 2, two experiments on the WMPPG aiming at BP prediction using four-wavelength PPG signals are conducted in this paper. Trial 2 was used as a comparison experiment to expand the ACNN part of the deep learning model into four side-by-side branches. The structure of each branch is the same as that of the input single-channel PPG signal, and all use RGB three-channel inputs. Thanks to the four ACNN branches, the model can input one to four channels of PPG at the same time. Then, the features extracted from the CNNs of the four branches are spliced, and the spliced features are input into the Bi-LSTM network in the next part. In Trial 3, only the 3-channel input of the CNN part is converted to a 12-channel input to process the previously mentioned 12-channel data obtained by fusing the four-wavelength PPG signals.

Subsequently, this paper conducted a comparative experiment between multi-wavelength PPG signals and single-wavelength BP prediction. It evaluated the effect of using four-wavelength PPG signals to predict BP, and the results are shown in Table 2. Table 2 shows that for the signals that were more challenging to predict SBP with, the results of Trial 2 improved the *R*^2^ from 0.89 to 0.94, the MAE from 3.57 mmHg to 1.88 mmHg, and the RMSE from 7.72 mmHg to 5.0 mmHg. The results of Trial 3 compared with those of the Trial 2 experiments showed that the *R*^2^ improved from 0.94 to 0.95, MAE from 1.88 mmHg to 1.67 mmHg, and RMSE from 5.66 mmHg to 5.28 mmHg compared with the results of Trial 2.

In order to further compare the BP prediction performance of Trial 2 and Trial 3, the British Hypertension Society (BHS) Programs [23] were used in this paper to evaluate the two experiments separately. According to the British Hypertension Society’s recommended rating criteria for blood pressure measurement equipment, grade A is given if 60% of measurements are differing from the true value by less than 5 mmHg, 85% of measurements are differing from the true value by less than 10 mmHg, and 95% of measurements are differing from the true value by less than 15 mmHg. The percentage of instruments with smaller prediction errors is definitely smaller than the percentage with larger prediction errors. Therefore, we can see in Table 3 that the percentage of prediction errors less than 5 mmHg, the percentage of prediction errors less than 10 mmHg, and the percentage of prediction errors less than 15 mmHg increase in that order. The specific evaluation results are shown in Table 3 and Table 4. Regarding SBP and DBP, both Trial 2 and Trial 3 achieved the A-level scores of the BHS index. Regarding predicting SBP, 94% of the Trial 3 data had a prediction error of less than 5 mmHg, while 89% of the Trial 2 data had a prediction error of less than 5 mmHg. Regarding predicting DBP, the proportion of prediction errors less than 5 mmHg in Trial 3 is even higher than 95%, while the ratio of prediction errors less than 5 mmHg taken in Trial 2 is 92%.

And, to further demonstrate the advantages of validating Trial 3 in predicting blood pressure, we show the Bland–Altman plots of the Trial 3 and Channel 4 experiments with the best performance in single-wavelength PPG. The Bland–Altman plot is an effective tool for assessing the consistency of two measures, favored for its intuitive graphical representation and low requirement for assumptions about data distribution. The plot makes it straightforward to identify a systematic bias and assess the level of consistency by showing the difference in the mean values of the two measurements and the consistency bounds of the difference. In addition, it reveals trends in differences with measurements and allows for the clinically meaningful assessment of the data. The Bland–Altman plot is also suitable for reproducibility assessment and subgroup analysis, enhancing its ability to analyze data under different conditions. Thanks to the support of a wide range of statistical softwares (Python 3.7, matplotlib 3.3.3), producing and analyzing Bland–Altman plots is fast and easy, making them widely used in medicine, biostatistics, and other scientific fields. As shown in Figure 4, the 95% confidence intervals of SBP and DBP for Trial 3 are in the range of [−10.129, 10.560] and [−4.052, 5.474]. The mean difference between the predicted and true values of SBP and DBP are 0.216 and 0.711. This indicates that Trial 3 can predict blood pressure with a high accuracy.

## 5. Discussion

The PPG signal, which can be easily collected using a photoelectric sensor placed on the skin’s surface, is widely used in physiological monitoring, health diagnosis, medical monitoring, and other fields. However, studies utilizing multi-wavelength PPG signals to estimate BP are relatively rare. In this study, we explored the feasibility and effectiveness of fusing four-channel PPG data through theoretical and experimental analyses and proposed the ACNN-BiLSTM model for BP prediction. We compared the performance of BP prediction using different single-wavelength PPG signals and using four-wavelength PPG signals. It is shown that the BP prediction model based on the ACNN-BiLSTM model with four-wavelength PPG signals has a higher accuracy and generalization ability.

### 5.1. Comparison of BP Prediction Using Single-Wavelength PPG Signals

In terms of predicting SBP, through Table 1, we can see that in the case of BP prediction using single-wavelength PPG signals, the best results were achieved using Channel 4 PPG signals for BP prediction which reached the AAMI criterion. This may be because Channel 4 has a wavelength of 940 nm, which can irradiate to a deeper depth than the rest of the channels and can acquire richer physiological information. This is precisely because more information about vascular reflexes and BP changes can be obtained. This advantage makes the scheme using Channel 4 achieve excellent performance in BP prediction. In addition, the experimental results also reveal the limitations exhibited by the other three channels. The wavelengths of these channels may not be able to reach the deeper tissues, as Channel 4 channel does, and therefore, do not fully capture the subtle differences in vascular reflexes and BP changes. Lacking this vital information, these channels exhibit relatively low levels of accuracy and reliability in BP prediction.

In terms of DBP prediction, there was no significant difference in DBP prediction using the PPG signals of the four channels, and all met the AAMI criteria. However, the PPG signal using Channel4 still showed a slight advantage. This is due to the more straightforward prediction of DBP relative to SBP. DBP requires less physiological information in the prediction process, so even if some of the channels are illuminated at shallow depths and do not have access to deeper physiological information under the skin, the DBP can still be predicted well by relying only on the physiological information from the more superficial layers of the skin surface. This result also demonstrates that the prediction of DBP relative to the prediction of SBP relies less on deeper physiological information. In contrast, the prediction of SBP requires more comprehensive and accurate physiologic information to achieve better results.

### 5.2. Advantages of Using Multi-Wavelength PPG Signals over Single Wavelengths

According to the observations in Table 1 and Table 2, BP prediction using four-wavelength PPG signals can significantly improve the accuracy of BP prediction compared to single-wavelength PPG signals. This is because when using four-wavelength PPG signals for BP prediction, more physiological information can be obtained to improve the accuracy of BP prediction compared to single-wavelength PPG signals. The four-wavelength PPG signal contains different wavelengths, each corresponding to a different tissue depth. In this way, by comprehensively analyzing the PPG signals of different channels, the prediction model can catch the characteristics of BP changes more thoroughly and accurately. Meanwhile, the advantage of using multi-wavelength PPG signals is that more physiological information about the blood vessels under the skin can be obtained. PPG signals of different wavelengths can be absorbed and reflected in tissues at different depths, providing a more comprehensive range of vascular reflexes and hemodynamic features. This integrated multichannel approach helps compensate for missing information or limitations with a single channel. In addition, using a four-wavelength PPG signal allows for better handling of interfering factors. Since different wavelengths have different absorption and scattering properties of light, the effect of some interfering signals on the prediction can be eliminated by comparing PPG signals of different wavelengths. By combining data from multiple channels, the prediction model can understand and analyze BP changes more comprehensively, thus improving the accuracy of BP prediction.

In Trial 3, better BP prediction can be achieved by the fused 12-channel array of four-wavelength PPG signals being directly input into the network model compared to Trial 2. This is because, through the fused array, the network model can simultaneously analyze and understand the relatedness and interactions between different channels. On the one hand, by fusing the four-wavelength PPG signals into a 12-channel array, the network model can more comprehensively access physiological information. Each channel represents a different depth of tissue information so that the whole 12-channel array can provide more prosperous and diverse features of BP changes. In this way, the prediction model can more accurately learn and understand the relatedness between different channels, thus improving the accuracy of BP prediction.

On the other hand, only the Bi-LSTM part of Trial 2 started to process the PPG signals of the four channels simultaneously. In contrast, Trial 3 was simultaneously processing four channels of data in both the CNN and the Bi-LSTM parts. This means that Trial 3 utilizes all channels of data more comprehensively and consistently throughout the prediction process. This unified processing may help improve the network model’s understanding and learning of the relatedness between the different channel signals, improving BP prediction.

### 5.3. Performance Comparison with Previous Studies Using Only PPG Waveforms for BP Estimation

This study presents a method for predicting BP using a four-wavelength PPG signal fusion method designed to predict BP by exploiting potential information between wavelengths. The fused four-wavelength data can be used to build BP estimation models that are fast, simple, robust, and do not require manual feature engineering to capture rapid, intermittent BP changes.

Table 5 compares our proposed method with PPG-based BP estimation studies published in the last few years, including the estimation dataset used, the length of the PPG signal used, and the model, mainly containing results from ME, SD, or MAE. Fair quantitative comparisons are difficult because of the differences in datasets, method implementation, and validation procedures among the studies. In addition, comparisons should be made, considering the trade-offs between performance, speed, and computational complexity metrics. Following this, we discuss the advantages offered by our proposed method.

### 5.4. Advantages and Disadvantages

This study aims to compare the effectiveness of BP prediction using PPG signals acquired at four different wavelengths and to provide a reference for wearable device manufacturers. At the same time, we propose a novel method of fusing multi-wavelength PPG signals and inputting them into a network model for BP prediction, significantly improving prediction accuracy. However, it should be pointed out that the performance of the neural network model relies heavily on a large amount of high-quality, noise-free PPG signal data, and the quality of the PPG signal is also affected by the acquisition environment. In this paper, the PPG signals were acquired in a quiet environment with suitable temperature and humidity, and the subjects were sedentary. Therefore, the feasibility and accuracy of this BP prediction method still need to be further evaluated for those cases where the acquisition environment is poor, or the subjects are in an active state.

In the current study, the validation of the model was limited to a relatively small group of hypertensive patients, which does limit our ability to assess the validity and generalizability of the model to a wider range of hypertensive patients. A hypertensive subgroup of only 66 participants may not fully capture the diversity of different hypertensive stages, ages, genders, lifestyles, and other variables that may influence model performance. To overcome this limitation, future studies should expand the sample size to include a larger number of hypertensive patients, which would allow for a more comprehensive assessment of the model’s performance in predicting and treating hypertension.

## 6. Conclusions

In this study, we propose a new scheme of fusing multiple wavelengths of PPGs and feeding them into a network model for BP prediction. Meanwhile, we verified the performance of different wavelength PPG signals and the simultaneous use of four-wavelength PPG signals in BP prediction. Through experimental analysis, especially in the prediction of SBP, the four-wavelength PPG signal data fusion designed in this paper can significantly improve the performance of the model for BP prediction after being input into the model. In this way, we can fully utilize the four-wavelength PPG data to provide better features for subsequent tasks. This study is not only of practical significance for developing wearable devices but it is also an essential contribution to BP prediction technology. Our findings provide valuable insights for future development in related fields and offer new possibilities for more accurate BP monitoring.

## Figures and Tables

**Figure 1 bioengineering-11-00306-f001:**
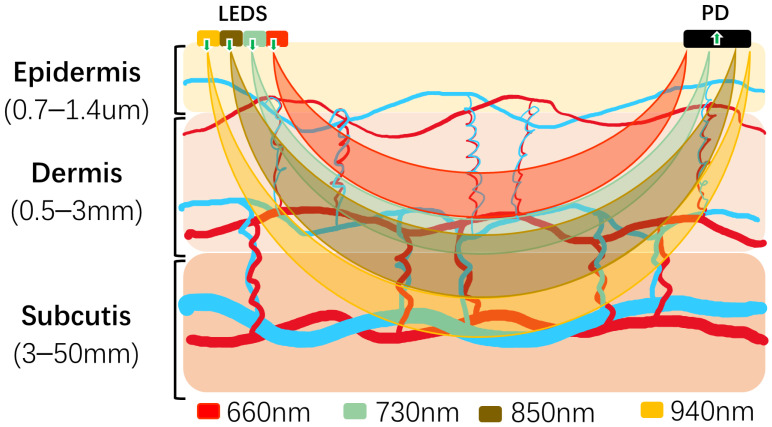
Schematic of reflective MWPPG collection.

**Figure 2 bioengineering-11-00306-f002:**
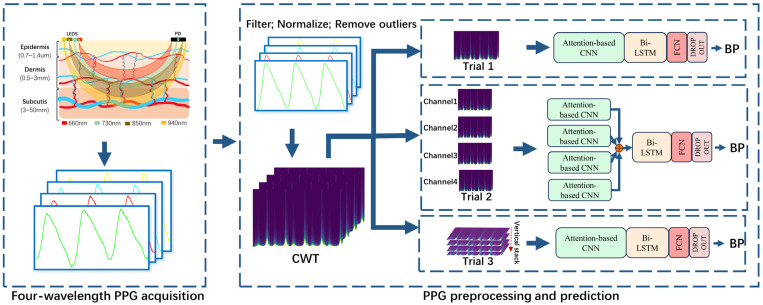
Schematic of reflective MWPPG collection.

**Figure 3 bioengineering-11-00306-f003:**
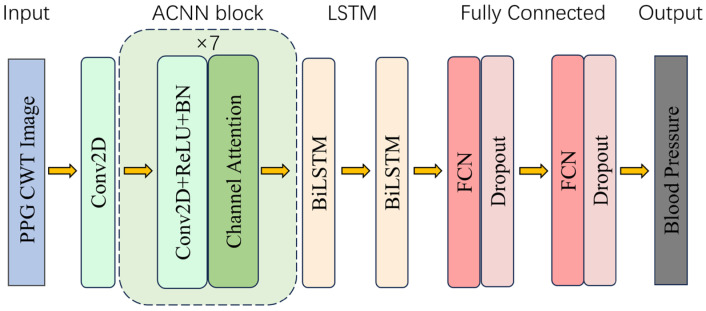
Schematic structure of the ACNN-BiLSTM BP prediction model.

**Figure 4 bioengineering-11-00306-f004:**
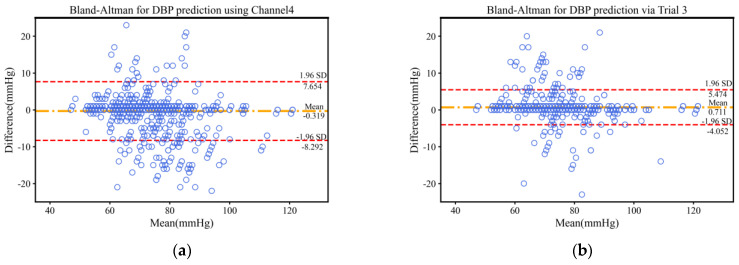
Bland−Altman plots for SBP (**top**) and DBP (**bottom**). The middle dotted line indicates the mean error between the predicted and true values. The upper and lower dotted lines indicate the upper and lower limits of the 95% consistency bounds. (**a**) Bland–Altman for DBP prediction using Channel 4; (**b**) Bland–Altman for DBP prediction via Trial 3; (**c**) Bland–Altman for SBP prediction using Channel 4; (**d**) Bland–Altman for SBP prediction via Trial 3.

**Table 1 bioengineering-11-00306-t001:** A comparison of the BP prediction performance using different channels’ data in Trial 1. Channel 1 is the result of BP prediction using Channel 1 (660 nm) data; Channel 2 is the result of BP prediction using Channel 2 (730 nm) data; Channel 3 is the result of BP prediction using Channel 3 (850 nm) data; Channel 4 is the result of BP prediction using Channel 4 (940 nm) data.

	SBP (mmHg)					DBP (mmHg)				
	*R* ^2^	*ME* ± *SD*	*MAE*	*RMSE*	AAMI	*R* ^2^	*ME* ± *SD*	*MAE*	*RMSE*	AAMI
Channel 1	0.85	0.58 ± 8.72	3.44	8.74	No	0.88	1.18 ± 4.40	2.29	4.55	Yes
Channel 2	0.82	0.01 ± 9.78	4.17	9.78	No	0.90	0.21 ± 4.15	1.66	4.15	Yes
Channel 3	0.87	0.89 ± 8.16	3.18	8.21	No	0.87	−0.66 ± 4.58	2.16	4.62	Yes
Channel 4	0.89	0.90 ± 7.66	3.57	7.72	Yes	0.90	−0.32 ± 4.07	1.70	4.08	Yes

**Table 2 bioengineering-11-00306-t002:** Comparison of the performance of the two feature fusion methods for predicting BP.

	SBP (mmHg)					DBP (mmHg)				
	*R* ^2^	*ME* ± *SD*	*MAE*	*RMSE*	AAMI	*R* ^2^	*ME* ± *SD*	*MAE*	*RMSE*	AAMI
Trial 2	0.94	−0.36 ± 5.65	1.88	5.66	Yes	0.93	0.67 ± 3.29	1.46	3.35	Yes
Trial 3	0.95	0.22 ± 5.28	1.67	5.28	Yes	0.96	0.71 ± 2.43	1.15	2.53	Yes

**Table 3 bioengineering-11-00306-t003:** Comparison of the two feature fusion methods on BHS schemes in predicting SBP.

Experience	Probability (%)		
	≤5 mmHg	≤10 mmHg	≤15 mmHg
Trial 2	89	94	96
Trial 3	94	96	97
Grade A	60	85	95
Grade B	50	75	90
Grade C	40	65	85

**Table 4 bioengineering-11-00306-t004:** Comparison of the two feature fusion methods on BHS schemes in predicting DBP.

Experience	Probability (%)		
	≤5 mmHg	≤10 mmHg	≤15 mmHg
Trial 2	92	97	98
Trial 3	95	98	99
Grade A	60	85	95
Grade B	50	75	90
Grade C	40	65	85

**Table 5 bioengineering-11-00306-t005:** Performance comparison with previous studies.

Reference	Dataset	Signal Length	Model	Performance	
				SBP (mmHg)	DBP (mmHg)
Ali et al. (2023) [15]	MIMIC-II	\	LSTM-ANN	3.39 ± 5.47 (ME ± SD)	1.79 ± 3.72 (ME ± SD)
Zhou et al. (2023) [24]	MIMIC-II	1024 sampling points	MultiResUNet3+	5.81 (MAE)	4.05 (MAE)
Wang et al. (2022) [25]	UCI	2 s	AlexNet	−0.00± 8.46 (ME ± SD)	−0.04 ± 5.36 (ME ± SD)
Leitner et al. (2022) [26]	MIMIC-III	5 s	CRNN-Transfer	3.52 (MAE)	2.20 (MAE)
Rong et al. (2021) [12]	UCI	680 sampling points	MTFF	5.59 (MAE)	4.48 (MAE)
El-Hajj et al. (2021) [27]	MIMIC-II	7 s	Attention-based RNN	2.58 (MAE)	1.26 (MAE)
Mahardika T et al. (2023) [28]	MIMIC-III	5 s	CNN-LSTM	3.79 ± 7.89 (MAE ± SD)	2.89 ± 5.34 (MAE ± SD)
De Oliveira et al. (2024) [29]	MIMIC-III	7 s	NeuBP	5.02 (MAE)	3.11 (MAE)
Bossavi et al. (2023) [30]	MIMIC-II	20 s	CNN	4.7 (MAE)	2.1 (MAE)
Attivissimo et al. (2023) [31]	MIMIC-III	2 s	XGBoost	5.67 (RMSE)	3.95 (RMSE)
This work	Self-prepared	5 s	ACNN-BILSTM	0.22 ± 5.28 (ME ± SD)	0.71 ± 2.43 (ME ± SD)

## Data Availability

The data used in this manuscript can be downloaded from this link https://doi.org/10.6084/m9.figshare.23283518.v1 (accessed on 19 September 2023).

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
