# Peer review of "ACNN-BiLSTM: A Deep Learning Approach for Continuous Noninvasive Blood Pressure Measurement Using Multi-Wavelength PPG Fusion"

_bioengineering, 2024, doi:10.3390/bioengineering11040306_

Round 1

Reviewer 1 Report

Comments and Suggestions for Authors

In the study ACNN-BiLSTM: A Deep Learning Approach for Continuous Noninvasive Blood Pressure Measurement Using Multi-wave length PPG Fusion” authors combine the continuous wavelet transform (CWT) into a multi-wavelength photoplethysmographic (MWPPG) fusion algorithm. After filtering MWPPG signals they applied the method of fusing the 4-channel CWT images into a 12-channel array for inputting into the deep learning model.

The measurement was performed on 162 volunteers out of which (only) 66 were hypertensive.  It would be good to repeat the measurements on only hypertensive volunteers.

How can you explain that the probability is better when ≤15mmHg (SBP/DBP)? Discuss this.

On the Figure 4 there are only Bland - Altman plots for SBP

There are typos mistake rows 305-308.

Reviewer 2 Report

Comments and Suggestions for Authors

In this paper, authors proposed A Deep Learning Approach for Continuous 2 Noninvasive Blood Pressure Measurement Using Multi-wave-3 length PPG Fusion.

The paper presents a promising method for non-invasive measurements. The study compared the performance of PPG 23 signals with different individual wavelengths and using a multi-wavelength PPG fusion method in 24 blood pressure prediction, assessed using mean absolute error (MAE), root mean square error 25 (RMSE) and AAMI-related criteria.

(1) Fig.1 has been taken from somewhere or has the authors drawn it by themselves ? Please cite reference if taken from other source.

(2) Keep the size of equations same.

(3) Font size in fig. 2 is quite small and visible to readers.

(4) In fig.4, why authors have only used Bland-Altman. any specific reasons ?

(5) Please expand the results section. The analysis should be in detail.

(6) In table-5, can you increase the number of similar parameter  comparisons ? Add more.

Reviewer 3 Report

Comments and Suggestions for Authors

The topic of the publication is very novel and important. Using wearable devices for BP control can significantly change the management of patients with Hypertension and save lives. The methodology of using 4-chanal PPG is described clearly. However, I have one question regarding the dataset for the training and test model. Usually, datasets are separated into three parts: training, validation, and testing. The authors show that the data set was divided into two groups: training and testing. How was the training model validated? Line 258-259

It will be interesting to check this model in patients with high blood pressure. It is the limitation of the study because only 66 participants had high BP. I recommend to add it to "Advantages and disadvantages.  

Minor errors

Line 336-337 Table 4 written by Rome numbers IV. Please correct.

Round 2

Reviewer 2 Report

Comments and Suggestions for Authors

None